# Doxorubicin·Hydrochloride/Cisplatin-Loaded Hydrogel/Nanosized (2-Hydroxypropyl)-Beta-Cyclodextrin Local Drug-Delivery System for Osteosarcoma Treatment In Vivo 

**DOI:** 10.3390/nano9121652

**Published:** 2019-11-21

**Authors:** Sun Jung Yoon, Young Jae Moon, Heung Jae Chun, Dae Hyeok Yang

**Affiliations:** 1Department of Orthopedic Surgery, Chonbuk National University Medical School, Research Institute of Clinical Medicine of Chonbuk National University-Biomedical Research Institute of Chonbuk National University Hospital, Jeonju 54907, Korea; sjyoon_kos@naver.com; 2Department of Biochemistry and Molecular Biology, Chonbuk National University Medical School, Jeonju 54896, Korea; yj-lunar@daum.net; 3Department of Medical Life Sciences, College of Medicine, The Catholic University of Korea, Seoul 06591, Korea; 4Institute of Cell and Tissue Engineering, College of Medicine, The Catholic University of Korea, Seoul 06591, Korea

**Keywords:** osteosarcoma, visible light-cured glycol chitosan hydrogel, (2-hydroxypropyl)-beta-cyclodextrin, doxorubicin·hydrochloride, cisplatin, local drug delivery system, cardiotoxicity, nephrotoxicity

## Abstract

Osteosarcoma (OSA) is a difficult cancer to treat due to its tendency for relapse and metastasis; advanced methods are therefore required for OSA treatment. In this study, we prepared a local drug-delivery system for OSA treatment based on doxorubicin·hydrochloride (DOX·HCl)/cisplatin (CP)-loaded visible light-cured glycol chitosan (GC) hydrogel/(2-hydroxypropyl)-beta-cyclodextrin (GDHCP), and compared its therapeutic efficiency with that of DOX·HCl- and CP-loaded GC hydrogels (GD and GHCP). Because of diffusion driven by concentration gradients in the swollen matrix, the three hydrogels showed sustained releases of DOX·HCl and CP over 7 days, along with initial 3-h bursts. Results of in vitro cell viability and in vivo animal testing revealed that GDHCP had a stronger anticancer effect than GD and GHCP even though there were no significant differences. Body weight measurement and histological evaluations demonstrated that the drug-loaded GC hydrogels had biocompatibility without cardiotoxicity or nephrotoxicity. These results suggested that GDHCP could be a good platform as a local drug-delivery system for clinical use in OSA treatment.

## 1. Introduction

Osteosarcoma (OSA) is a common primary bone cancer that frequently occurs in young people with bimodal peak incidence [1]. Despite advanced development of OSA theragnosis, cancer metastasis continues to occur; therefore, a 2-year survival rate is only 15% to 20% [2]. OSA is generally treated with anticancer drugs for approximately 10 weeks before surgery; this treatment is known as neoadjuvant chemotherapy. After surgery, the cancer is again treated with adjuvant chemotherapy for approximately one year. Surgery on its own improves the survival duration from 134 to 175 days, while adjuvant chemotherapy with a single anticancer drug improves the survival duration from 262 to 413 days, on average. As such, chemotherapy is among the most commonly used treatments for OSA.

Anticancer drugs for chemotherapy include doxorubicin (DOX), cisplatin (CP), epirubicin, ifosfamide, cyclophosphamide, eotposide, gemcitabine, and topotecan. These anticancer drugs improve the survival duration of patients with OSA; however, the cancer is still prone to relapse and metastasis. Additionally, the drugs sometimes induce multidrug resistance (MDR) [3,4]. Hence, new, advanced chemotherapeutic approaches are required for successful treatment of OSA [5].

Combination chemotherapy is more effective than single drug chemotherapy for neoplasia inhibition [6,7]. For example, compared with single anticancer drugs, combination chemotherapy using DOX and CP led to an average survival time of 540 days. Appropriate designs of drug-carrier systems are necessary for improving the performance of combination chemotherapy [8].

Among drug-carrier systems, nanomaterial-based systemic drug carriers require several characteristics to specifically deliver anticancer drugs to targeted sites [9]. These characteristics include shape, size and surface charge, which improve passive targeting, and ligand conjugation, which improves active targeting [9]. Although systemic drug-delivery systems are efficient, their therapeutic applications are sometimes limited by complicated preparation. Hydrogel-medicated local drug delivery systems are capable of directly delivering anticancer drugs to targeted sites without complicated procedures because they can be implanted near cancer tissues [10,11,12]. We have already reported that a visible light-cured injectable glycol chitosan (GC) hydrogel-based local drug-delivery system can be used as an anticancer drug carrier for treatment of several solid cancers [10,11,12]. Additionally, beta-cyclodextrin (β-CD) is used as a nanosized drug carrier for anticancer drugs because it has a cavity that is similar in size to the drugs [13,14]. Among β-CD derivatives, (2-hydroxypropyl)-beta-cyclodextrin (HP-β-CD) enhances the poor water solubility (1 mg/mL in water with warming) of CP through inclusion complex formation [15].

In this study, we prepared a doxorubicin·hydrochloride (DOX·HCl) and CP-loaded hydrogel/nanosized HP-β-CD drug delivery system for OSA treatment. In vitro and in vivo anticancer effects were evaluated using a KHOS/NP human OSA cell line and an OSA cancer-bearing xenograft mouse model, respectively.

## 2. Materials and Methods

### 2.1. Materials

GC (≥60% calculated by titration, crystalline, MW ≈ 585,000 g/mol) and glycidyl methacrylate (GM) were purchased from Sigma-Aldrich (St. Louis, MO, USA). Riboflavin 5′-monophosphate sodium salt (riboflavin; Santa Cruz Biotechnology, Inc., Santa Cruz, CA, USA) was used for photo-curing. Doxorubicin·hydrochloride (DOX·HCl; Tokyo Chemical Industry Co., Ltd., Tokyo, Japan) and cisplatin (CP; Sigma-Aldrich, St. Louis, MO, USA) were used for OSA therapy. (2-hydroxypropyl)-beta-cyclodextrin (HP-β-CD; St. Louis, MO, USA) was used to improve the water solubility of CP. Cellulose membrane (cut-off: 20 kDa) was purchased from Spectrum Laboratories Inc. (Rancho Dominguez, CA, USA). MG-63 and KHOS/NP human osteosarcoma cell lines were obtained from American Type Culture Collection (ATCC; Manassas, VA, USA). A cell counting kit-8 (CCK-8; Dojindo Molecular Technologies, Inc. Rockville, MD, USA) was used for in vitro cell viability assay. All chemicals were used as received.

### 2.2. HP-β-CD ((2-Hydroxypropyl)-Beta-Cyclodextrin) and Cisplatin (CP) Complex (HPCD/CP)

According to a previous report, an inclusion complex was formed between HP-β-CD and CP [15]. Briefly, CP (0.03 mmol, 5 mg) was added to a solution of HP-β-CD (0.03 mmol, 41.88 mg) in water (10 mL), and the mixture was sonicated at room temperature for 1 h. After passing through a filter with specific pore size 0.22 µM, the filtrate was lyophilized at −90 °C to obtain white powder.

### 2.3. Preparation of Doxorubicin·Hydrochloride (DOX·HCl) and/or HPCD/CP-Loaded Visible Light-Cured Glycol Chitosan (GC) hydrogel (DOX·HCl-Loaded (GD), CP-Loaded (GHCP) and GDHCP)

The methacrylic group was conjugated to GC for photo-curing [10,11,12]. GC (0.003 mmol, 1.5 g) was dissolved in water (500 mL). GM (0.05 mmol, 7 mg) was added to aqueous GC solution, and then adjusted to pH 9. The mixture was reacted at room temperature for 2 days. After neutralization, the reactant was dialyzed (cut-off: 20 kDa) in water until it was purified. The clear solution was lyophilized at −90 °C and stored at −20 °C before use. Riboflavin (12 µM), DOX·HCl (2 mg), and/or HPCD/CP (CP: 2 mg) were added to an 1 mL of aqueous methacrylic GC solution (1 *w*/*v*%), and dispersed homogeneously. The mixture was irradiated for 10 s using blue light (430–485 nm, 2100 mW/cm^2^, light-emitting diode curing light, Foshan Keyuan Medical Equipment Co., Ltd., Guangdong, China) for hydrogel formation. The swelling ratio was then calculated by the ratio of swollen weight measured at each determined time interval (0, 1, 2, 3, 4, 5, 6, 7, 14, 21 and 28 days) to the initial hydrogel weight in PBS (pH 7.4) at 37 °C.

### 2.4. Release Test of DOX·HCl and CP

Three kinds of hydrogels with specific amounts of DOX·HCl (2 mg/mL) and/or CP (2 mg/mL mg), GD, GHCP and GDHCP, were added to a cellulose membrane (cut-off: 3500 g/mol). After immersion in a 50 mL Falcon tube filled with 37 mL (PBS, pH 7.4), the mixtures were continuously shaken at 100 rpm at 37 °C. At predetermined time intervals (1, 3, 6, 12, 24, 48, 72, 96, 120, 144, and 168 h), a specific volume of efflux (3 mL) was extracted, and the same of volume PBS was then added. Release tests of DOX·HCl and CP were conducted using a UV-visible spectrophotometer at 480 and 310 nm, respectively. Standard curves of DOX·HCl and CP were prepared using several dilutions of each drug.

### 2.5. In Vitro Cell Viability

KHOS/NP (5 × 10^3^ cells/well) and MG-63 cells (5 × 10^3^ cells/well) were seeded on 24-well plates, respectively. Specific volumes of GD (1 mL), GHCP (1 mL; CP: 50 µg/mL) and GDHCP (1 mL; DOX·HCl: 50 µg/mL and CP: 50 µg/mL)-loaded trans-well insert was added to the 24-well plates. Then, KHOS/NP and MG-63 cells were cultured with α-minimum essential medium Eagle (MEM; Life Technologies, Carlsbad, CA, USA) supplemented with 10% fetal bovine serum (Life Technologies, Carlsbad, CA, USA) and Dulbecco’s minimum essential medium high glucose (DMEM) supplemented with 10% fetal bovine serum (Life Technologies, Carlsbad, CA, USA) and 1% penicillin-streptomycin Life Technologies, Carlsbad, CA, USA), respectively. After seeding, control (cell-seeded well plate) and hydrogel-treated cells were incubated at 37 °C for 1, 3, 5 and 7 days under 5% CO_2_ conditions. At the determined time intervals, the sample-treated cells were washed with PBS (pH 7.4). CCK-8 (10 µL) was added to each of sample-treated cells, after which, the cells were incubated for an additional 4 h and measured with a microplate reader at 450 nm.

### 2.6. Osteosarcoma (OSA) Animal Model

The study protocol was approved by the Institutional Animal Care and Use Committee of Chonbuk National University (Permit No: CBNU 2019-011). Six-week-old male BALB/c nude mice (Orient Bio, Seongnam, Gyonggi-Do, Korea) were used for the OSA model; they were quarantined for one week prior to experimentation. The mice had free access to food and water and were kept in a room with controlled humidity (50%) and temperature (22 °C) on a 12-h light/dark cycle. Mice at seven weeks of age were anesthetized with ketamine (100 mg/kg) and xylazine (5 mg/kg), and a surgical incision was made in the left knee joint area. After incision of the patellar tendon, a needle was inserted through the tibia plateau. KHOS/NP cells (1.0 × 10^5^ cells) (ATCC, Manassas, VA, USA) in 10 μL PBS were injected into the marrow space using a Hamilton syringe. Four weeks after cancer cell inoculation, the mice were randomly distributed into five groups of four mice each. Thereafter, a specific volume (100 µL) of normal saline, GD (DOX·HCl: 2 mg/kg), GHCP (CP: 2 mg/kg), and GDHCP (DOX·HCl: 2 mg/kg and CP: 2 mg/kg) were injected around the cancer area once a week for 4 weeks. The cancer volume was calculated using the following formula: V = 0.5 × longest diameter × (shortest diameter)^2^. Body weight and cancer size were measured weekly for four weeks. The mice were euthanized four weeks later, and histological evaluations were performed.

### 2.7. Histological Evaluations

Cancer, heart, and kidney tissues were extracted from each mouse and fixed using 4 *v*/*v*% of formaldehyde for one day. The tissues were embedded in paraffin after being dehydrated with a series of gradient ethanol concentrations. The blocked tissues were sliced to a thickness of 3 µm and stained with H&E. The stained slides were observed using slide scanner (Pannoramic MIDI; 3DHISTECH Ltd., Budapest, Hungary) and panoramic viewer (Version 1.15.3; Pannoramic MIDI; 3DHISTECH Ltd., Budapest, Hungary) program.

### 2.8. Statistical Analysis

All quantitative data were expressed as mean ± standard deviation. A statistical analysis was performed using one-way analysis of variance (ANOVA) with SPSS software (SPSS Inc., Chicago, IL, USA). A value of * *p* < 0.05 was considered statistically significant.

## 3. Results

### 3.1. Swelling Ratio

The swelling ratios of GC, GD, GHCP and GDHCP, shown in Figure 1, were measured at 37 °C for four weeks. The swelling ratios increased gradually for 7 days and remained constant thereafter. Regardless of the addition of drugs, no significant difference in the swelling ratio was observed. This result may be attributed to the three-dimensional network of the hydrogel, which caused the matrix to be swollen from penetration by water molecules.

### 3.2. Release Behavior of DOX·HCl and/or CP

The release behaviors of the drugs in GD, GHCP and GDHCP were examined at 37 °C for seven days because the hydrogel was injected once a week for 4 weeks and the results are shown in Figure 2. The three hydrogels exhibited an initial burst release for 3 h and controlled releases in a sustained manner thereafter. The initial burst was dependent on the unique characteristics of the polymer matrix; water-soluble forms of DOX·HCl and HPCD/CP near the matrix were rapidly released while drugs in the matrix bulk were released slowly by concentration gradient-motivated diffusion.

### 3.3. In Vitro Anticancer Effect

Figure 3 shows the in vitro cell proliferation rates of MG-63 and KHOS/NP cells cultured on control, GD, GHCP and GDHCP at 37 °C for 1, 3, 5 and 7 days. In both cancer cells, control showed a gradual increase in cell proliferation rate for seven days, while the proliferation rate in hydrogels gradually decreased. Additionally, on day 7, GDHCP had 1.06- and 1.05-fold lower MG-63 cell proliferation rate and 1.06- and 1.03-fold lower KHOS/NP cell proliferation rate than GD and GHCP due to the cocktail effect of the two drugs.

### 3.4. In Vivo Anticancer Effect

Figure 4 shows the gross appearances and cancer volumes of GC, GD, GHCP and GDHCP-treated mice compared with those of the control. As shown in the gross appearances of Figure 4, the drug-loaded samples had smaller cancer sizes than the control and GC at 4 weeks after drug injection. The cancer volumes were measured for 4 weeks (the cancer volume results of Figure 4), and a gradual increase in cancer volume was observed in the control and GC-treated mice. The cancer volumes in the drug-loaded samples decreased gradually for 4 weeks. In addition, the cancer volumes of GD, GHCP and GDHCP-treated mice at 4 weeks were 164 ± 68 mm^3^, 178 ± 89 mm^3^ and 121 ± 26 mm^3^, respectively. Furthermore, GDHCP resulted in 3.73, 3.31, 1.36 and 1.47 fold lower cancer volumes than control, GC, GD and GHCP.

### 3.5. Body Weight

Figure 5 shows the body weights of the control, GC, GD, GHCP and GDHCP-treated mice over 4 weeks. The body weights of the control and GC-treated mice after four weeks were 1.12 and 1.15 fold larger than before the treatment, which can be attributed to cancer tissue growth. In GD, GHCP and GDHCP-treated mice, a marginal difference in body weight was observed throughout the test period. This finding indicates that the anticancer effect of the drugs stopped the cancer volume from increasing. GDHCP exhibited a superior anticancer effect even though the cancer volume was not significant when compared to the GD and GHCP-treated mice.

### 3.6. Histological Evaluations

Figure 6, Figure 7 and Figure 8 show the H&E-stained images of cancer, heart and kidney tissues in cancer-bearing mice treated with drug-loaded GC hydrogels for 4 weeks, as compared to the control. Figure 6 shows the H&E-stained images of cancer tissues for all mouse groups. Dense cancer cells were observed in the H&E-stained images of control and GC-treated mice. The drug-loaded samples induced cancer tissue necrosis, and GDHCP led to extended necrosis because of the synergistic effect of DOX·HCl and CP. The H&E-stained images of heart tissues for all mouse groups are shown in Figure 7. Compared with heart tissues of the control group, DOX·HCl-loaded samples did not induce abnormal cellular/tissue responses, indicating that there was no cardiotoxicity of GD and GDHCP. Figure 8 shows H&E-stained images of kidney tissues for all mouse groups. No abnormal cellular/tissue responses were observed in the control or in GHCP- and GDHCP-treated mice. These findings demonstrated that the GC hydrogel can be used as a local drug-delivery system for anticancer drugs to targeted sites. Furthermore, GDHCP noticeably induced cancer tissue necrosis without cardiotoxicity or nephrotoxicity.

## 4. Discussion

The present study investigated the feasibility of a hydrogel/nanosized HP-β-CD drug-delivery system based on visible-light-cured, injectable GC hydrogel combination DOX·HCl and HPCD/CP complex for in vitro and in vivo OSA treatment. Although chemotherapy is the most frequently used treatment for OSA, it has an approximately 30% failure rate due to MDR [16,17]. Drug resistance is caused by various mechanisms, including decreased drug accumulation by P-glycoprotein, apoptosis inhibition by Bcl-2, p53 and miRNAs, up-regulation of nuclear factor kappa B activity, blocking between drugs and DNA Topo II, cell detoxification by GSTP1, and enhancement of DNA repair by external RNA controls consortium (ERCC) or apurinic/apyrimidinic endonuclease 1 (APE1) [18,19].

Combined chemotherapy is known to overcome MDR in OSA [20]. Among anticancer drugs, DOX is standard among leading drugs for OSA chemotherapy [21]. A potent anticancer drug that is often used for OSA treatment, CP triggers cell apoptosis by binding to DNA [22]. In addition to MDR, rapid elimination of anticancer drugs in the blood stream by non-specific protein binding results in low therapeutic efficacy and undesired side effects [23].

Drawbacks such as MDR and poor blood circulation may be diminished by the hydrogel-based local drug-delivery system, because the proposed system can deliver anticancer drugs to cancer tissue without being processed in the blood stream, and can minimize MDR with combined chemotherapy [10,11,12]. Among hydrogel systems, an injectable hydrogel is valuable, both because it can directly deliver anticancer drugs near cancer tissues by local injection and because it has a potential for drug loading in the matrices. In our previous studies, the visible light-cured GC hydrogel system was found to be a potent local drug-delivery carrier because of its injectability and the fact that it allows anticancer drugs to be controllably released in a sustained manner [10,11,12].

The release behavior of drugs in hydrogels depends mainly on gradient concentration diffusion [24]. This diffusion leads to both initial bursts and controlled release due to the drug distribution in hydrogels. The two release behaviors should be attributed to the diffusion of drugs distributed near the hydrogel surface and bulk, respectively. Additionally, the drug diffusion is closely related to the swelling of porous hydrogels. We have previously reported that the GC hydrogel has porous structure together with interconnectivity [10,11,12]. These structural properties resulted in in vivo anticancer effects against solid cancers when the GC hydrogel was used as the local anticancer drug-delivery system [10,11,12]. Therefore, it is assumed that GDHCP has a superior anticancer effect against osteosarcoma in vitro and in vivo (Figure 3, Figure 4, Figure 5, Figure 6, Figure 7 and Figure 8).

Despite strong anticancer effects, DOX and CP have cardiotoxic and nephrotoxic effects [25,26,27], respectively, that decrease their therapeutic efficiencies. DOX causes cumulative and dose-dependent cardiotoxicity, including abnormal changes to the structure and function of myocardium, severe cardiomyopathy, and congestive heart failure, resulting in cardiac transplants and death in many patients [25,26]. CP is finally cleared by the kidneys through glomerular filtration and tubular secretion; therefore, the drug molecules are filtered out when the concentration in the kidney exceeds the concentration in the blood. The filtered drugs are accumulated in renal parenchymal cells, leading to nephrotoxicity [28].

Because of serious DOX and CP toxicity, histological evaluations of heart and kidney tissues were performed using H&E stained images. As shown in Figure 7 and Figure 8, we confirmed the biocompatibility of GDHCP against the heart and kidney tissues. The results of this study suggest that GDHCP has clinical potential for osteosarcoma treatment (Figure 9).

## 5. Conclusions

This study investigated DOX·HCl and CP-loaded injectable visible light-cured GC hydrogel/ nanosized HP-β-CD drug-delivery systems for in vitro and in vivo OSA treatment. Along with the swollen properties of the hydrogel, the nanosized HP-β-CD used for improving the poor water solubility of CP exhibited controlled release behavior of DOX·HCl and/or CP in a sustained manner, resulting in potential anticancer effects on OSA. Furthermore, drug-loaded GC hydrogels did not affect the heart or kidney. Through an in vivo animal study, although significant differences in the tumor volumes on GD-, GHCP-, and GDHCP-treated mice were not found, GDHCP improved the antitumor effect in OSA treatment as compared with GD and GHCP. These results suggested the feasibility of GDHCP for the local cancer treatment of OSA.

GDHCP did not affect the heart or kidney. Our results suggest that this injectable GDHCP system has the potential for clinical use in OSA treatment.

## Figures and Tables

**Figure 1 nanomaterials-09-01652-f001:**
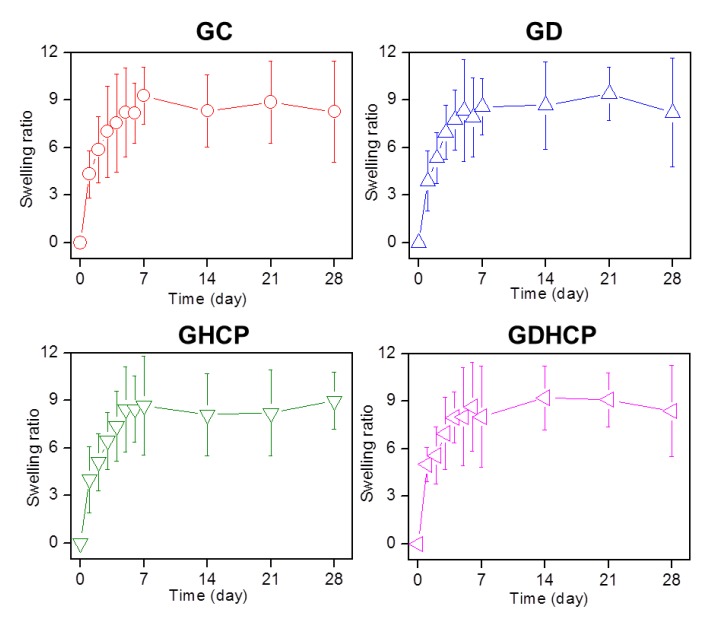
Swelling ratios of cured glycol chitosan (GC) hydrogel, DOX·HCl-loaded GC hydrogel (GD), CP-loaded GC hydrogel (GHCP) and GDHCP measured at day 0, 1, 2, 3, 4, 5, 6, 7, 14, 21, and 28 days. At the determined intervals, the hydrogels immersed in PBS (pH 7.4) at 37 °C were extracted, dried and weighted. This experiment was performed three times.

**Figure 2 nanomaterials-09-01652-f002:**
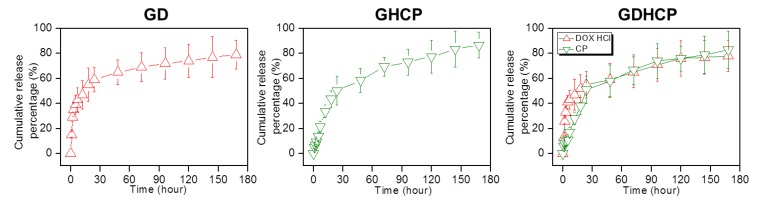
Cumulative release percentages of DOX·HCl and/or CP in GD, GHCP and GDHCP measured at day 1, 3, 6, 12, 24, 48, 72, 96, 120, 144, and 168 h. This experiment was performed three times in PBS (pH 7.4) at 37 °C.

**Figure 3 nanomaterials-09-01652-f003:**
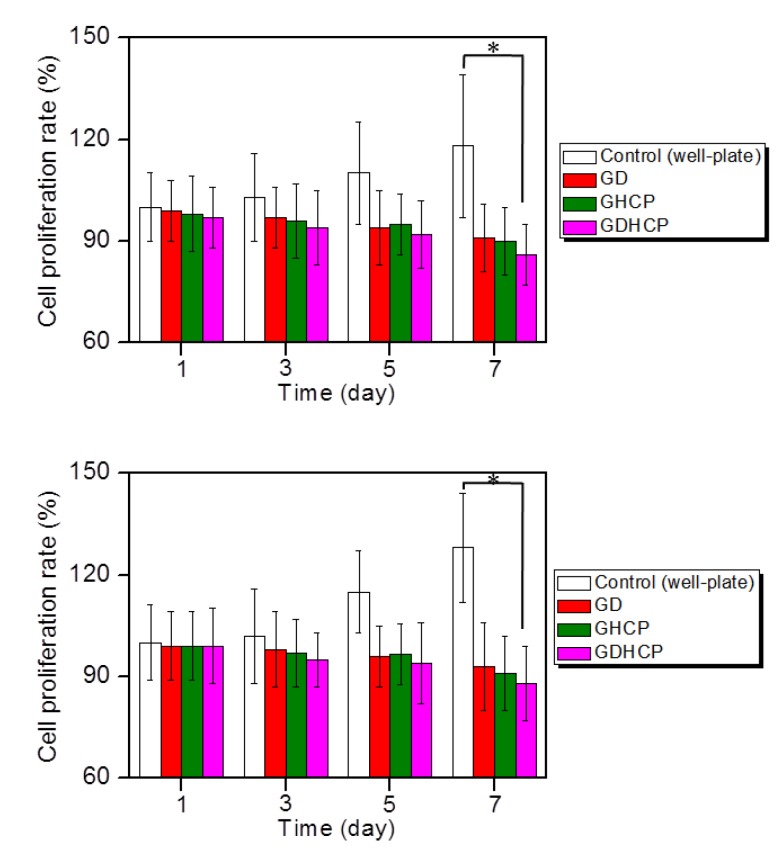
Proliferation rates of (top) MG-63 and (down) KHOS/NP cells cultured with control, GD, GHCP and GDHCP (DOX·HCl: 2 mg/kg and CP: 2 mg/kg against each hydrogel) at 37 °C for 1, 3, 5 and 7 days under 5% CO_2_ conditions. Practical treated amounts of DOX·HCl and CP were 50 ng because the average body weight was 25 mg. This experiment was performed three times (* *p* < 0.05). The significance was compared with the control.

**Figure 4 nanomaterials-09-01652-f004:**
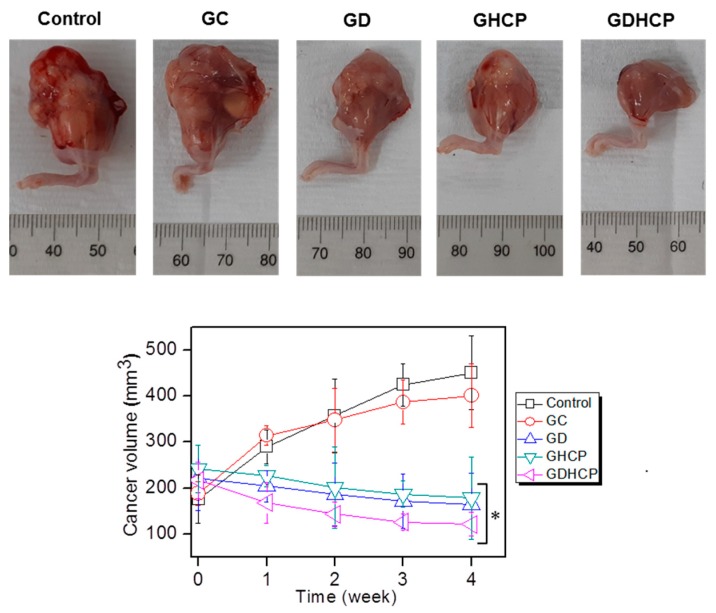
(**Top**) Gross appearances of cancer bearing to the femoral lesion of control, GC, GD, GHCP and GDHCP observed at 4 weeks after intratumoral injection. The injection was performed once a week for 4 weeks. (**Bottom**) Cancer volumes in control, GC-, GD-, GHCP- and GDHCP-treated mice (# *p* < 0.05). The significance was compared with the control.

**Figure 5 nanomaterials-09-01652-f005:**
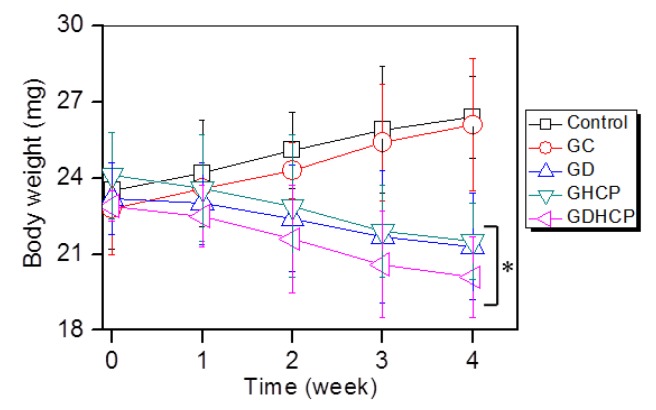
Body weights of control, GC-, GD-, GHCP- and GDHCP-treated mice measured at 1, 2, 3 and 4 weeks. This experiment was performed three times (* *p* < 0.05). The significance was compared with the control.

**Figure 6 nanomaterials-09-01652-f006:**
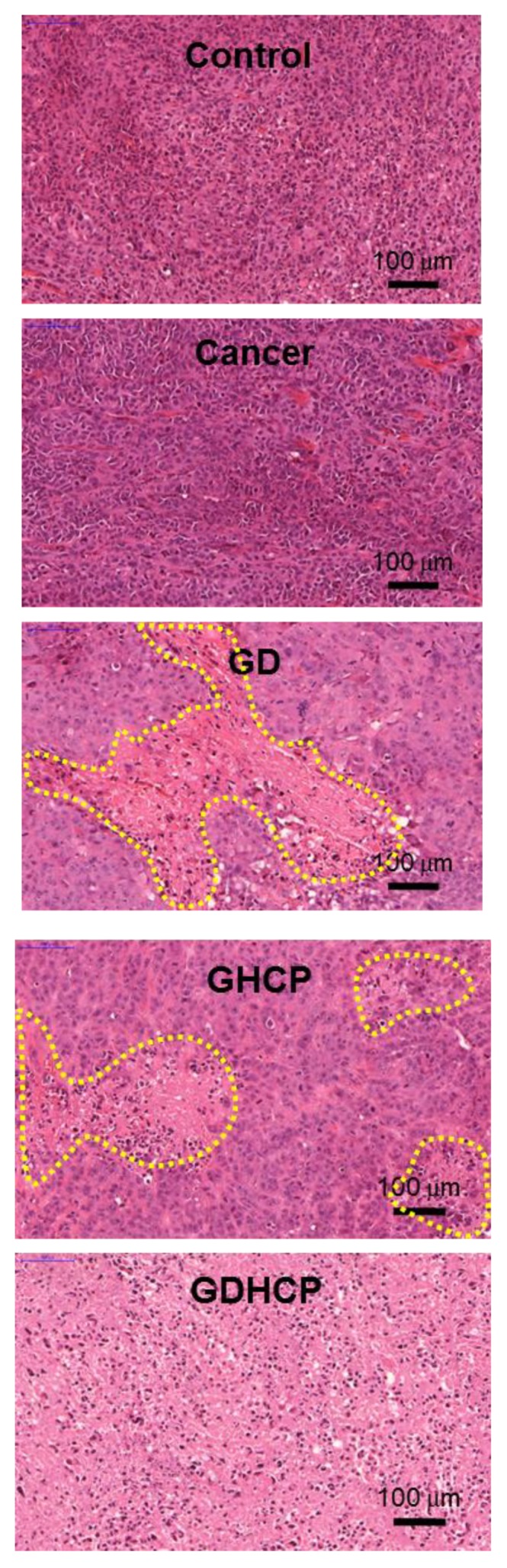
H&E stained slides of cancer tissues extracted from control, cancer bearing, GD-, GHCP- and GDHCP-treated mice after 4 weeks. The slides were observed at 20×. The scale bars indicate 100 µm. The yellow dotted lines indicated necrotic area.

**Figure 7 nanomaterials-09-01652-f007:**
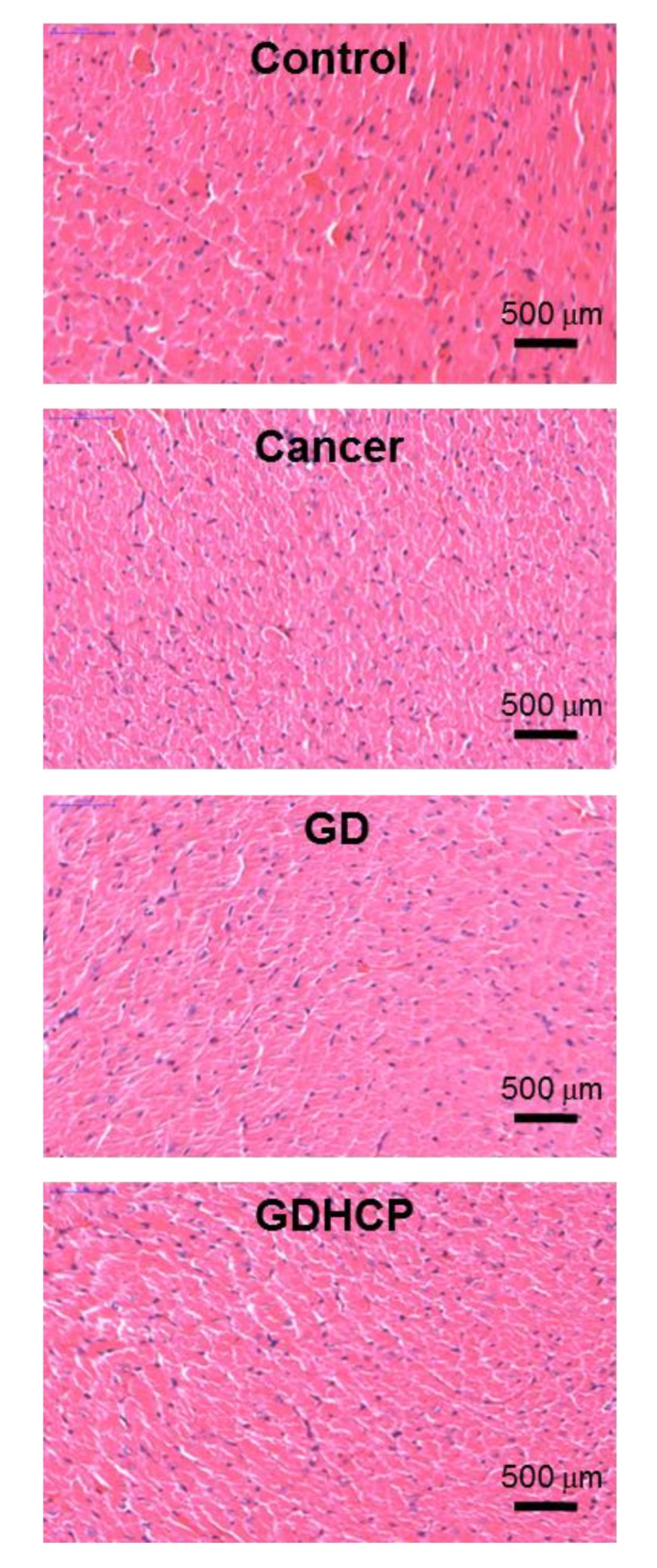
H&E stained slides of heart tissues extracted from control, cancer bearing, GD- and GDHCP-treated mice after 4 weeks. The slides were observed at 4×. The scale bars indicate 500 µm.

**Figure 8 nanomaterials-09-01652-f008:**
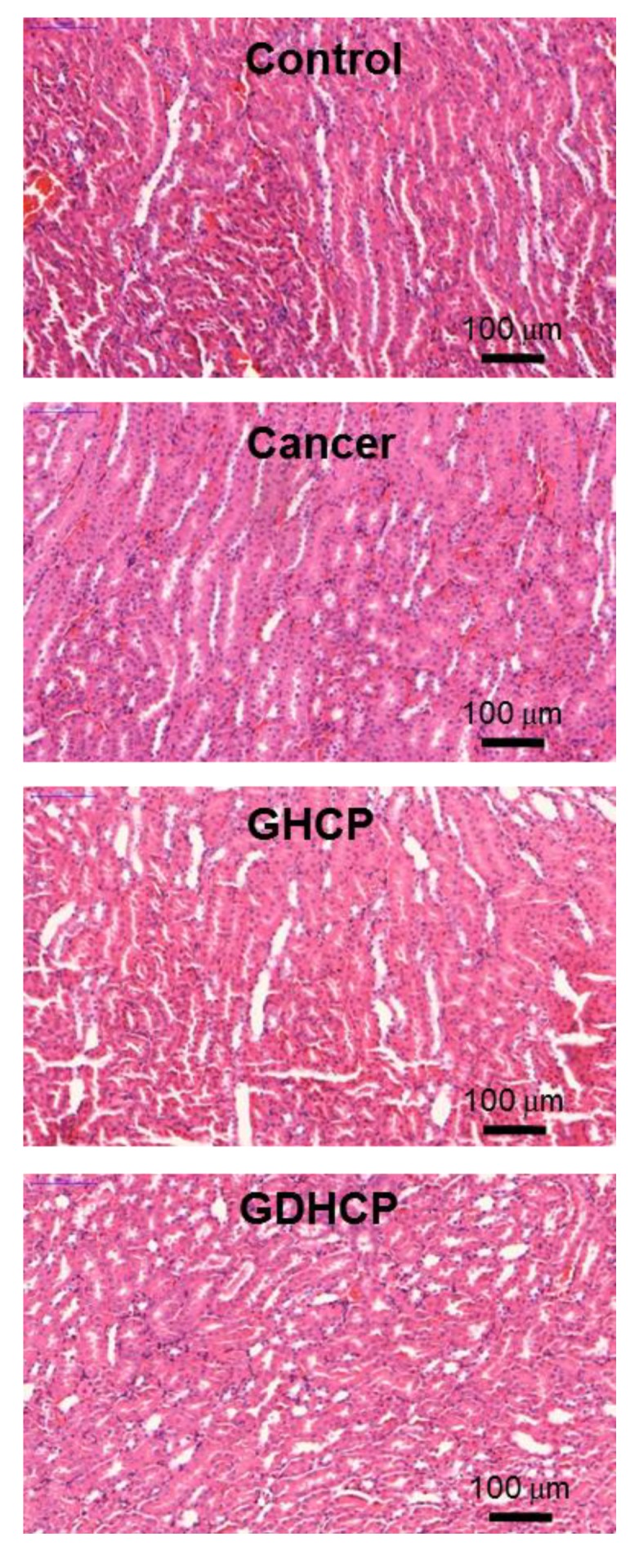
H&E stained slides of heart tissues extracted from control, cancer bearing, GHCP- and GDHCP-treated mice after 4 weeks. The slides were observed at 20×. The scale bars indicate 100 µm.

**Figure 9 nanomaterials-09-01652-f009:**
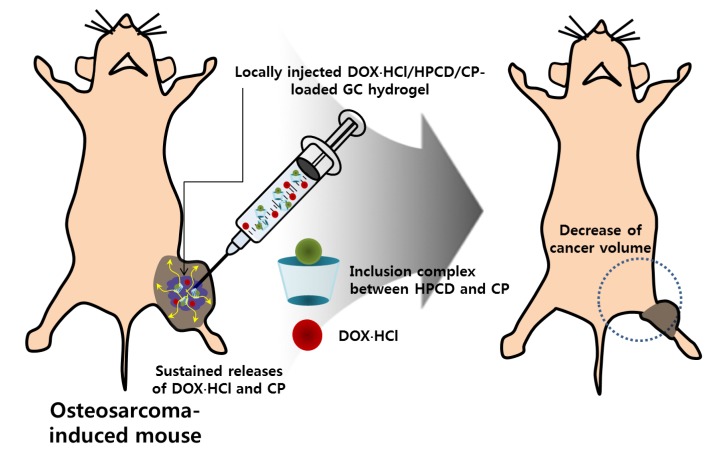
In vivo anticancer effect of GDHCP hydrogel on osteosarcoma. The hydrogel was locally injected into the cancer tissue once a week for 4 weeks.

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
