# Peer review of "Doxorubicin·Hydrochloride/Cisplatin-Loaded Hydrogel/Nanosized (2-Hydroxypropyl)-Beta-Cyclodextrin Local Drug-Delivery System for Osteosarcoma Treatment In Vivo"

_nanomaterials, 2019, doi:10.3390/nano9121652_

Round 1

Reviewer 1 Report

The authors investigated the development of various hydrogels as a local drug delivery system for osteosarcoma treatment. From the results of this study, the authors showed the anticancer activity of hydrogels from in vitro to in vivo. Although the results showed the strong anticancer effect over the control group (GD or control), there is no significant difference between each hydrogel such as GD, GHCP, and GDHCP. Thus, the result sentence by the authors “GDHCP could be a good platform as a local drug delivery system” is not conclusive.

To make the authors' conclusion effective, they need to show the significant difference among those hydrogels (GD, GHCP, and GDHCP). Otherwise, it is a huge challenge to publish this study.

Author Response

The authors investigated the development of various hydrogels as a local drug delivery system for osteosarcoma treatment. From the results of this study, the authors showed the anticancer activity of hydrogels from in vitro to in vivo. Although the results showed the strong anticancer effect over the control group (GD or control), there is no significant difference between each hydrogel such as GD, GHCP, and GDHCP. Thus, the result sentence by the authors “GDHCP could be a good platform as a local drug delivery system” is not conclusive.

Q1) To make the authors' conclusion effective, they need to show the significant difference among those hydrogels (GD, GHCP, and GDHCP). Otherwise, it is a huge challenge to publish this study

A1) Thank you for your valuable comment. As you noted, your thought is correct, and therefore, the authors revised the result of GDHCP treatment and expressed in the Abstract and Conclusion.

Reviewer 2 Report

The manuscript entitled “Doxorubicin-Hydrochloride/Cisplatin-loaded hydrogel/nanosized (2-Hydroxypropyl-Beta-Cyclodextrin Local Drug Delivery System for Osteosarcoma Treatment In Vivo” describes a new nanosized hydrogel for a local delivery of chemotherapeutics (i.e. doxorubicin plus cisplatin) in order to improve the efficacy and decrease the non-specific toxicity of the osteosarcoma (OSA) treatments.

The Authors demonstrate a superior “in vivo” efficacy of this new drug formulation with respect to other single-agent formulations (i.e. GD and GHCP); moreover, the local controlled release eliminates the cardiac and renal toxicities due to doxorubicin and cisplatin, respectively.

This hydrogel is also endowed with good swelling and releasing properties which foster a slowly drug discharge after an initial fast-releasing phase.

The topics discussed in this manuscript are very appealing because OSA treatment is challenging due to the high percentage of relapsing patients and the high resistance of this tumor to chemotherapeutics.

However, although the questions addressed by the paper are very interesting, some issues need to be clarified before publication:

Authors suggest applying this nano-formulation for the local delivery of chemotherapeutics at the tumor site. One of the main problems of OSA is the development of pulmonary metastasis and when they are radiological detected the 5 years survival rates are only 30%. So, if the tumor spreading is very wide and characterized by small spotted tumor masses, how do the Authors envisage to apply their new hydrogel to fight this metastatic malignancy?

PARAGRAPH 3.3. In vitro antitumor effect How did Authors decide the dose of drugs (i.e. DOX-HCl 2 mg/mL and CP 2 mg/mL) used in this experiment? Did they also perform other experiments in order to study the dose/efficacy curves? In my opinion, the efficacy of doxorubicin and cisplatin alone and in association should be shown in Figure 3 to compare the drug cytotoxicity with respect the nano-formulations. Moreover, the Y-scale of Figure 3 should be rescaled to better appreciate the differences among the curves; the symbols can be explained in the legend of Figure 3. Finally, it should be important to show the “in vitro” efficacy on other OSA cell lines.

PARAGRAPH 3.4. In vivo anticancer effect The Y-scale of Figure 4 should be rescaled to better appreciate the differences among the curves; the symbols can be explained in the legend of Figure 4. Furthermore, it should be important to show in the text the statistical analysis, p value, among the different “in vivo” treatments (i.e. GD, GHCP, and GDHCP). Why did Authors decide to use the dose of DOX-HCl 2 mg/kg and CP 2 mg/kg? Did you test higher doses? Are higher doses toxic at the cardiac and renal level?

PARAGRAPH 3.5. Body weight The Authors show in Figure 5 that there is a significant difference in the body weight between the untreated and the drug-loaded nano-formulation treated mice. How can they exclude that the decreasing of the weight is partially due to the toxicity of the treatments? The Y-scale of Figure 5 should be rescaled to better appreciate the differences among the curves; the symbols can be explained in the legend of Figure 5.

Author Response

The manuscript entitled “Doxorubicin-Hydrochloride/Cisplatin-loaded hydrogel/nanosized (2-Hydroxypropyl-Beta-Cyclodextrin Local Drug Delivery System for Osteosarcoma Treatment In Vivo” describes a new nanosized hydrogel for a local delivery of chemotherapeutics (i.e. doxorubicin plus cisplatin) in order to improve the efficacy and decrease the non-specific toxicity of the osteosarcoma (OSA) treatments.

The Authors demonstrate a superior “in vivo” efficacy of this new drug formulation with respect to other single-agent formulations (i.e. GD and GHCP); moreover, the local controlled release eliminates the cardiac and renal toxicities due to doxorubicin and cisplatin, respectively.

This hydrogel is also endowed with good swelling and releasing properties which foster a slowly drug discharge after an initial fast-releasing phase.

The topics discussed in this manuscript are very appealing because OSA treatment is challenging due to the high percentage of relapsing patients and the high resistance of this tumor to chemotherapeutics.

However, although the questions addressed by the paper are very interesting, some issues need to be clarified before publication:

Q1) Authors suggest applying this nano-formulation for the local delivery of chemotherapeutics at the tumor site. One of the main problems of OSA is the development of pulmonary metastasis and when they are radiological detected the 5 years survival rates are only 30%. So, if the tumor spreading is very wide and characterized by small spotted tumor masses, how do the Authors envisage to apply their new hydrogel to fight this metastatic malignancy?

A1) Thank you for your valuable comment. Currently used methods for identifying metastatic malignant cancers have been confirmed using advanced techniques such as PET-CT. Therefore, in the case of currently used chemotherapeutic methods (oral administration, intravenous injection, etc.), the anticancer drugs reach the cancer tissue by systemic circulation, and various side effects occur when this process is performed. On the other hand, our system can be used in most organ because it is possible to inject minimally though the syringe into the area where cancer tissue has developed. Thus, the present system can be applied to multiple organs in which metastatic malignant cancers are found and can be as effective system compared to the currently used methods.

Q2) PARAGRAPH 3.3. In vitro antitumor effect How did Authors decide the dose of drugs (i.e. DOX-HCl 2 mg/mL and CP 2 mg/mL) used in this experiment? Did they also perform other experiments in order to study the dose/efficacy curves? In my opinion, the efficacy of doxorubicin and cisplatin alone and in association should be shown in Figure 3 to compare the drug cytotoxicity with respect the nano-formulations. Moreover, the Y-scale of Figure 3 should be rescaled to better appreciate the differences among the curves; the symbols can be explained in the legend of Figure 3. Finally, it should be important to show the “in vitro” efficacy on other OSA cell lines.

A2) Thank you for your valuable comment. The authors are sorry for the reviewer about wrong expression of the usage because the 2 mg/mL must to be replaced with 2 mg/kg. The dosage of anticancer drugs is generally determined by the body surface area. The body surface area is calculated by the Mosteller formula, and in this experiment, the practical body surface area of the used mice was calculated to yield approximately 2 mg/kg. Therefore, the 2 mg/kg represents a substantial clinical dosage amount.

Our experiment was carried out based on the determination of anticancer drug dosage using body surface area. This may not make dose/efficacy curves unnecessary. The authors sincerely hope the reviewer understands our research direction.

In addition, the reviewer suggested the evaluation of doxorubicin and cisplatin alone on in vitro antitumor effect. However, the doxorubicion is HCl salt form, which is water-soluble. This water soluble doxorubicin can easily penetrate across cell membrane, resulting in a superior antitumor effect. We have to focus on the inconsistency between in vitro and in vivo results, because anticancer drugs are generally used in vivo and the drugs meet some obstacles in blood vessels. Therefore, the suggested experiment will not match the direction of our research.

As you suggested, the Y-scale was revised and the symbols were already explained in the legend.

Also, the authors carried out the in vitro antitumor effect using another OSA cell (MG-63 cell) and the methods and results were added in the manuscript.

Q3) PARAGRAPH 3.4. In vivo anticancer effect The Y-scale of Figure 4 should be rescaled to better appreciate the differences among the curves; the symbols can be explained in the legend of Figure 4. Furthermore, it should be important to show in the text the statistical analysis, p value, among the different “in vivo” treatments (i.e. GD, GHCP, and GDHCP). Why did Authors decide to use the dose of DOX-HCl 2 mg/kg and CP 2 mg/kg? Did you test higher doses? Are higher doses toxic at the cardiac and renal level?

A3) Thank you for your valuable comment. The dosage of anticancer drugs is generally determined by the body surface area. The body surface area is calculated by the Mosteller formula, and in this experiment, the practical body surface area of the used mice was calculated to yield approximately 2 mg/kg. Therefore, the 2 mg/kg represents a substantial clinical dosage amount.

Our experiment was carried out based on the determination of anticancer drug dosage using body surface area. Therefore, in vitro studies at higher doses will not match the direction of our research. The authors sincerely hope the reviewer understands our research direction.

Q4) PARAGRAPH 3.5.Body weight The Authors show in Figure 5 that there is a significant difference in the body weight between the untreated and the drug-loaded nano-formulation treated mice. How can they exclude that the decreasing of the weight is partially due to the toxicity of the treatments? The Y-scale of Figure 5 should be rescaled to better appreciate the differences among the curves; the symbols can be explained in the legend of Figure 5.

A4) Thank you for your valuable comment. The authors evaluated the toxicity of kidney and heart to identify the toxicities of the hydrogel samples that may be caused by the treatment. This is believed to mean that the weight is decreased by reducing the size of cancer tissues treated with the hydrogel samples, rather than the toxicity of the sample themselves.

The Y-scale was rescaled as you noted and the symbols were already explained in the legend.

Round 2

Reviewer 1 Report

The authors elaborated on the reviewer’s comment. Thus, it will be suited for this publication without further change.

Author Response

Q1) The authors elaborated on the reviewer’s comment. Thus, it will be suited for this publication without further change.

A1) We appreciate your valuable comment.

Reviewer 2 Report

I have only a minor observation.

Pag 3 In vitro Viability- The Authors indicated the drug concentrations using mg/mL; when they described the results of the same IN VITRO experiments (i.e. Figure 3), they used a different concentration mg/kg.

In my opinion the concentration mg/mL is Ok for the IN VITRO assays .

The concentration mg/kg should only be used for IN VIVO experiments.

Author Response

Q1) Pag 3 In vitro Viability- The Authors indicated the drug concentrations using mg/mL; when they described the results of the same IN VITRO experiments (i.e. Figure 3), they used a different concentration mg/kg.

In my opinion the concentration mg/mL is Ok for the IN VITRO assays .

The concentration mg/kg should only be used for IN VIVO experiments.

A1) Thank you for your valuable comment and we apologize our mistake. The amounts of DOX.HCl and CP were determined by their initial injection amounts.

That is, the average weight of mice used were 25 g. Therefore, the practical amounts of DOX.HCl and CP used for intial injection were 50 ug and 50 ug, respectively. Since the in vitro cell viability test was performed for 1 week, we used the determined amounts of DOX.HCl and CP and expressed in Secion 2.5.